# Ipragliflozin Additively Ameliorates Non-Alcoholic Fatty Liver Disease in Patients with Type 2 Diabetes Controlled with Metformin and Pioglitazone: A 24-Week Randomized Controlled Trial

**DOI:** 10.3390/jcm9010259

**Published:** 2020-01-18

**Authors:** Eugene Han, Yong-ho Lee, Byung-Wan Lee, Eun Seok Kang, Bong-Soo Cha

**Affiliations:** 1Division of Endocrinology, Department of Internal Medicine, Keimyung University School of Medicine, Daegu 42601, Korea; jinsjins85@gmail.com; 2Division of Endocrinology, Department of Internal Medicine, Yonsei University College of Medicine, Seoul 03722, Korea; yholee@yuhs.ac (Y.-h.L.); bwanlee@yuhs.ac (B.-W.L.); edgo@yuhs.ac (E.S.K.); 3Institue of Endocrine Research, Yonsei University College of Medicine, Seoul 03722, Korea

**Keywords:** non-alcoholic fatty liver disease, obesity, type 2 diabetes mellitus, sodium/glucose cotransporter 2 inhibitor

## Abstract

Despite the benefits of pioglitazone in the treatment of non-alcoholic fatty liver disease (NAFLD), many treated patients continue to experience disease progression. We aimed to investigate the additive effect of ipragliflozin on NAFLD in patients with type 2 diabetes treated with metformin and pioglitazone. In this 24-week randomized controlled trial, 44 patients with type 2 diabetes and comorbid NAFLD were either randomized to receive 50 mg/day of ipragliflozin as an add-on treatment (*n* = 29) or maintained on metformin and pioglitazone (*n* = 15). The fatty burden was assessed using the fatty liver index, NAFLD liver fat score, and controlled attenuation parameter (CAP). Changes in fat and muscle depots were measured by dual-energy x-ray absorptiometry and abdominal computed tomography scans. The enrolled patients were relatively controlled (mean baseline glycated hemoglobin of 6.6% ± 0.6%) and centrally obese (mean waist circumference of 101.6 ± 10.9 cm). At week 24, patients in the ipragliflozin add-on group exhibited reduced hepatic fat content (fatty liver index: −9.8 ± 1.9, *p* = 0.002; NAFLD liver fat score: −0.5 ± 0.2, *p* = 0.049; CAP: −8.2 ± 7.8 dB/m^2^, *p* = 0.133). Ipragliflozin add-on therapy also reduced whole-body visceral fat and the ratio of visceral to subcutaneous fat (change in whole-body visceral fat: −69.6 ± 21.5 g; change in abdominal visceral fat: −26.2 ± 3.7 cm^2^; abdominal visceral to subcutaneous fat ratio: −0.15 ± 0.04; all *p* < 0.05). In conclusion, ipragliflozin treatment significantly ameliorates liver steatosis and reduces excessive fat in euglycemic patients with type 2 diabetes and NAFLD taking metformin and pioglitazone.

## 1. Introduction

As obesity and type 2 diabetes (T2D) emerge as comorbid endemic diseases, management strategies have shifted to account for obesity and T2D-related complications. Non-alcoholic fatty liver disease (NAFLD) is an obesity-related disease that is both a complication of and a risk factor for type 2 diabetes [1,2,3]. Traditionally, thiazolidinediones (TZDs) are the first-line treatment for patients with comorbid T2D and NAFLD [4]. TZDs are peroxisome proliferator-activated receptor gamma agonists that improve glucose tolerance and insulin sensitivity and enhance insulin action in the muscle, liver, and heart as well as in adipose tissue [5,6,7]. TZDs in patients with comorbid T2D and NAFLD have been shown to improve liver enzymes and hepatic histology [8]. Hepatic steatosis reduction independent of glycemic control has also been observed [5]. TZDs are ineffective in some patients with T2D and their use is associated with an increased risk for edema, fluid retention, and weight gain, particularly subcutaneous fat gain [9]. Moreover, the response rate to TZD treatment is relatively low (approximately 50%) in NAFLD [8,10,11].

Sodium/glucose cotransporter 2 (SGLT2) inhibitors have received significant attention due to their unique mechanism of action, whereby they inhibit glucose re-absorption independent of insulin [12]. Recent studies have reported the benefits of SGLT2 inhibitors in cardiovascular diseases [13,14] and renal protection [15,16,17]. In addition, it has been demonstrated that SGLT2 inhibitors reduce visceral fat and attenuate inflammation [18]. Despite several studies in animal models and in patients with T2D [19], an understanding of how SGLT2 inhibitors affect NAFLD remains limited.

We hypothesized that ipragliflozin, an SGLT2 inhibitor, would reduce excessive fat in patients with comorbid T2D and NAFLD, and therefore serve as a candidate therapy for patients with NAFLD who are unresponsive to metformin and TZD combination therapy. The aim of this study, therefore, was to investigate changes in fat depots and the severity of NAFLD in patients taking ipragliflozin for 24 weeks in addition to metformin and pioglitazone compared to treatment with metformin and pioglitazone only.

## 2. Methods

### 2.1. Study Design

A randomized, 24-week, controlled, parallel, open-label study was conducted at the Severance Hospital, a tertiary university hospital in Seoul, Korea, from April 2016 to June 2017. We enrolled patients between 20 and 75 years of age with well-controlled type 2 diabetes (glycated hemoglobin (HbA1c) levels ≤ 9.5% or 80 mmol/mol) and a body mass index (BMI) ≥ 23 kg/m^2^ (cut-off for overweight according to the Asia-Pacific definition). Patients were eligible for the study if they were receiving metformin and pioglitazone combination therapy for at least 8 weeks and naive to treatment with any SGLT2 inhibitors. All patients exhibited NAFLD, as diagnosed by abdominal ultrasound and confirmed by radiologic specialists. The exclusion criteria for this study were: (1) diagnosis of type 1 diabetes, gestational diabetes, or any diabetes diagnosis other than type 2 diabetes; (2) history of addiction to alcohol, heavy alcohol consumption (≥210 g/week for men or ≥140 g/week for women), or patients with aspartate aminotransferase (AST), alanine aminotransferase (ALT), or bilirubin levels more than three times the upper normal limit; (3) other causes of liver disease (e.g., active viral or autoimmune hepatitis), liver cirrhosis, or hepatocellular carcinoma; (4) estimated glomerular filtration rate (eGFR) < 60 mL/min/1.73 m^2^; (5) medication associated with fatty liver disease (e.g., amiodarone, methotrexate, tamoxifen, or valproate) or weight loss; and (6) pregnant or nursing women. This study complied with the Declaration of Helsinki and Good Clinical Practice guidelines. The study protocol received ethical approval by the institutional review board at the Yonsei University College of Medicine (4-2015-1115). All subjects provided written informed consent. This study is described further at https://clinicaltrials.gov/ (NCT02875821).

Eligible patients were randomly assigned in a 1:2 ratio using a computer-generated randomization sequence to either receive 50 mg/day of ipragliflozin as an add-on to metformin and pioglitazone combination treatment or were maintained on metformin and pioglitazone combination therapy. No dosage adjustments were made to any of the study medications in either study arm during the study period. All patients received diet and exercise counseling at the beginning of the study and were reminded at each study visit to follow their recommended plan.

### 2.2. Laboratory and Imaging Studies

All individuals underwent physical examination and clinical laboratory tests after an overnight (≥8 h) fasting period at baseline. Liver fat content was assessed via the controlled attenuation parameter (CAP). Briefly, the CAP measures ultrasonic attenuations at 3.5 MHz using signals acquired by transient liver elastography (FibroScan^®^; Echosens, Paris, France). The final CAP value was the median of each individual CAP value using the same valid measurements [5,20]. We also analyzed each patient’s fatty liver index [21] and NAFLD liver fat score [22]. The equations are described in Appendix A.

Whole-body fat distribution and muscle mass were measured via dual-energy x-ray absorptiometry (DXA). An abdominal fat computed tomography (CT) scan (Tomoscan 350; Philips, Mahwah, NJ, USA) was performed to measure the abdominal subcutaneous fat area (SFA), abdominal visceral fat area (VFA), and ratio of VFA to SFA. Serial DXA examinations were performed in each patient using the same machine (QDR-4500W; Hologic, Bedford, MA, USA) throughout the study. The abdominal fat depot was measured by obtaining a single cross-sectional CT image of a 3-mm thick slice at the level of the L4–L5 interspace while the subject was in a supine position. The VFA and SFA were electronically calculated using the TeraRecon Aquarius software (Aquaris iNtuition Ver.4.4.6 TeraRecon, Foster City, CA, USA) over an attenuation range of 2150 to 250 Hounsfield units, as described previously [23,24]; the results herein are reported in cm^2^. The VFA was determined by measuring the intra-abdominal cavity at the internal aspect of the abdominal and oblique muscle walls surrounding the cavity and the posterior aspect of the vertebral body. The remaining fat interposed between the muscle and subcutaneous tissue was selected and calculated as the SFA. The ratio of VFA to SFA was calculated as VFA/(VFA + SFA).

Body weight, waist circumference, blood pressure, glycemic parameters (i.e., fasting plasma glucose (FPG) and HbA1c), lipids (i.e., total cholesterol, high-density lipoprotein cholesterol (HDL-C), low-density lipoprotein cholesterol (LDL-C), and triglycerides), and liver enzymes (i.e., AST, ALT, and gamma glutamyl transferase (γ-GT)) were measured at the start of the study and at weeks 12 and 24.

### 2.3. Outcomes

The primary outcome was the change in total visceral fat as measured by DXA after 24 weeks. The key secondary outcomes were changes in CAP, fatty liver index, and NAFLD fatty liver score. In addition, we compared changes in SFA, VFA, SFA/VFA ratio, glycemic parameters, lipid profile, and liver enzymes at week 24. The homeostasis model assessment of insulin resistance (HOMA-IR) and HOMA-β were quantified based on FPG and fasting insulin levels using the following calculations: HOMA-IR = FPG (mg/dL) × insulin (mIU/L)/405 and HOMA-β = (360 × insulin (mIU/L))/(FPG (mg/dL) − 63)%. The percent change (Δ) in each parameter was calculated as: (baseline value − 24-week value/baseline value) × 100 (%).

### 2.4. Statistical Analysis

The planned sample size was 45 subjects in a randomized 1:2 ratio (*n* = 15 for metformin + pioglitazone and *n* = 30 for metformin + pioglitazone + ipragliflozin), which was calculated a priori to have 90% power to detect a difference of 0.08 kg in visceral fat based on a standard deviation (SD) of 0.1 kg at α = 0.05 with a discontinuation rate of 10%. The data are presented as the mean ± SD for continuous variables and as the number or percent for categorical variables. We analyzed differences in participant characteristics between groups using paired t-tests for continuous variables and χ^2^ tests for categorical variables. The total cholesterol, triglyceride, HDL-C, LDL-C, AST, ALT, γ-GT, insulin, HOMA-IR, and HOMA-β values were not normally distributed; analyses, therefore, were performed using log-transformed data. We tested treatment differences in the primary and key secondary endpoints using analysis of covariance (ANCOVA) models with treatment and sex as fixed effects and baseline values as covariates. To evaluate the association among changes in body weight, VFA, and muscle mass in the ipragliflozin add-on group, we performed Pearson’s correlation analyses. A responder to ipragliflozin was defined as any individual who exhibited a decrease in body weight of more than 1.6 kg (median weight loss of the ipragliflozin group) after treatment. Multivariable-adjusted logistic regression analyses were performed to test the independent association between ipragliflozin response and other clinical factors. All statistical analyses were performed using IBM SPSS version 23.0 for Windows (IBM Corp., Armonk, NY, USA); *p* < 0.05 was considered statistically significant.

## 3. Results

### 3.1. Baseline Characteristics of the Study Population

In total, 55 patients were screened and 45 patients with comorbid type 2 diabetes and NAFLD were enrolled. Glycemic parameters were stable (mean FPG = 119.6 ± 20.9 mg/L and HbA1c = 6.6% ± 0.6%, 49.0 ± 7.1 mmol/mol), confirming the effectiveness of metformin + pioglitazone treatment in these patients (Table 1). Thirty subjects were randomly assigned to the ipragliflozin add-on group and 15 patients were assigned to the metformin + pioglitazone maintenance group. One patient withdrew consent during the study period; 44 patients completed the study through week 24 (Appendix A). The mean age of the patients was 53.9 ± 10.9 years, and 62.2% of the patients were male. The average time since diagnosis of type 2 diabetes was 9.4 ± 5.8 years, and 64.4% and 97.8% of patients had hypertension and dyslipidemia, respectively. The mean body weight was 83.3 ± 14.6 kg, and the mean BMI was 30.3 ± 4.6 kg/m^2^. Although all enrolled patients had NAFLD, their liver function test results were almost within normal limits (37.6 ± 39.4 IU/L, 27.7 ± 15.4 IU/L, and 32.3 ± 21.5 IU/L for γ-GT, AST, and ALT, respectively). The mean CAP was 306.3 ± 38.3 dB/m. The mean fatty liver index score was 29.9 ± 20.3 and the mean NAFLD liver fat score was −1.9 ± 1.4. The mean VFA-to-SFA ratio was 47.0 ± 11.8%. The mean values for total fat mass, total fat ratio, estimated visceral fat, and total muscle mass were 25.3 ± 7.8 kg, 29.8% ± 6.9%, 683.5 ± 173.4 g, and 56.3 ± 11.2 kg, respectively. Baseline characteristics were similar between the two groups.

### 3.2. Effect of Ipragliflozin Add-On Treatment on Body Weight and Fat Depots

Body weight, BMI, and waist circumference were significantly decreased in the ipragliflozin add-on group at 24 weeks compared to baseline (−1.6 ± 0.4 kg, −0.6 ± 0.1 kg/m^2^, and −3.2 ± 0.8 cm, respectively; all *p* < 0.05) (Table 2). Reductions in body weight and BMI were initially observed at 12 weeks and continued to decrease through 24 weeks (Figure 1A,B). The waist circumference was reduced in the ipragliflozin add-on group at week 24 only (adjusted mean difference of −2.71%, 95% CI = −4.65 to −0.78, *p* = 0.001) (Figure 1C).

A statistically significant reduction in total visceral fat was observed in the ipragliflozin add-on group at 24 weeks compared to the metformin + pioglitazone maintenance group (mean change in total visceral fat: −69.64 g (95% CI = −117.59 to −20.78 g) versus 17.83 g (95% CI = −49.58 to 85.23 g), ipragliflozin add-on versus metformin + pioglitazone maintenance groups, respectively). The VFA and VFA-to-SFA ratio were also decreased in the ipragliflozin add-on group (adjusted change in mean VFA: −26.57 cm^2^ (95% CI = −35.35 to −17.79 cm^2^) versus 7.79 cm^2^ (95% CI = −4.44 to 20.02 cm^2^); adjusted change in mean VFA-to-SFA ratio: −0.15% (95% CI = −0.23% to −0.07%) versus 0.12% (95% CI = 0.00% to 0.23%); ipragliflozin add-on versus metformin + pioglitazone maintenance group, respectively) (Figure 1D–F). SFA decreased in both groups, and total muscle mass decreased in the ipragliflozin group, but these changes were not statistically significant.

### 3.3. Effect of Ipragliflozin Treatment on NAFLD, Glycemic, and Lipid Parameters

The NAFLD surrogate marker, ALT, was significantly decreased in the ipragliflozin group (from 33.4 ± 25.1 IU/L to 25.6 ± 16.9 IU/L and from 31.1 ± 13.5 IU/L to 26.5 ± 11.8 IU/L in the ipragliflozin add-on and metformin + pioglitazone maintenance groups, respectively; *p* < 0.001) Similarly, γ-GT was significantly decreased in the ipragliflozin group (from 40.9 ± 47.8 IU/L to 29.7 ± 24.6 IU/L and from 31.8 ± 15.6 IU/L to 29.9 ± 13.8 IU/L in the ipragliflozin add-on and metformin + pioglitazone maintenance groups, respectively; *p* < 0.001) (Table 2). Both the fatty liver index and the NAFLD liver fat score were significantly reduced in the ipragliflozin add-on group (−9.80, 95% CI = −13.7 to −6.0; and −0.54, 95% CI = −0.86 to −0.23, respectively), but not in the metformin + pioglitazone maintenance group (1.05, 95% CI = −4.32 to 6.42; and 0.01, 95% = −0.44 to 0.44, respectively) (Figure 2). The adjusted mean change in CAP exhibited a non-significant decline in only the ipragliflozin add-on group (−7.98 dB/m; 95% CI = −9.96 to 33.2). FPG and HbA1c levels also were slightly decreased in the ipragliflozin add-on group, but no significant difference between the two groups was observed. In addition, HOMA-IR improved in the ipragliflozin add-on group (from 2.7 ± 1.8 to 2.2 ± 1.4). Patients in the metformin + pioglitazone maintenance group exhibited a greater reduction in total cholesterol and LDL-C, whereas patients in the ipragliflozin add-on group exhibited a non-significant elevation in both HDL-C and LDL-C despite a reduction in total cholesterol.

### 3.4. Correlation between Changes in Body Weight, VFA, and Muscle Mass Following Ipragliflozin Add-On Treatment

Body weight, visceral fat, and muscle mass were reduced in the ipragliflozin add-on group. We, therefore, explored the associations between changes in body weight, VFA, and muscle mass (Figure 3). The reduction in body weight had a significant moderate positive correlation with the change in VFA (Pearson’s correlation coefficient = 0.422, *p* = 0.023) and with the change in CAP (Pearson’s correlation coefficient = 0.505, *p* = 0.006). No correlations were observed between the VFA and change in muscle mass (Pearson’s correlation coefficient = 0.041, *p* = 0.833), the reduction in body weight and the change in SFA (Pearson’s correlation coefficient = 0.170, *p* = 0.377), or the reduction in body weight and change in VFA (Pearson’s correlation coefficient = −0.203, *p* = 0.292). We identified the ipragliflozin responders as subjects with body weight reduction (1.6 kg, median value). Ipragliflozin response group tended to have higher baseline CAP and muscle mass; however, there was no clinical factor associated with ipragliflozin response in multivariable-adjusted logistic regression analyses.

### 3.5. Safety≈

During the study period, 24 (61.4%) adverse events occurred. Eight patients (18.2%) exhibited symptoms of hypoglycemia, 11 patients (25.0%) reported renal and urinary disorders (e.g., polyuria and pollakiuria without progression to chronic kidney disease), and 3 patients (6.8%) had cystitis. No neoplasm development or extremity amputation occurred during the study period.

## 4. Discussion

In this open-label, randomized, controlled study, ipragliflozin as an add-on to metformin and pioglitazone in euglycemic patients with T2D and NAFLD was associated with decreased liver fat content and visceral fat depots at week 24. Although combination metformin and pioglitazone therapy is recommended for patients with NAFLD, some patients continue to experience NAFLD progression. Ipragliflozin add-on reduced body weight, waist circumference, and visceral fat, and also improved the fatty liver index, NAFLD liver fat score, and CAP. Moreover, the reduction in body weight correlated with changes in both abdominal visceral fat and CAP. The changes in abdominal visceral fat were not associated with changes in muscle mass. 

The current study has several strengths. First, although several studies have proposed that SGLT2 inhibitors lower visceral fat and provide cardiovascular protection in patients with type 2 diabetes [13,14,25], this study is, as far as we know, the first to investigate the influence of SGLT2 inhibitors on whole-body fat depots and NAFLD in patients with T2D on metformin and pioglitazone maintenance therapy. The safety and efficacy of SGLT2 inhibitors as an add-on therapy to metformin and pioglitazone have been established [26,27,28]. These prior studies, however, have focused on the efficacy of SGLT2 inhibitors on reducing blood glucose and body weight. Moreover, these prior studies were conducted in patients with T2D with uncontrolled hyperglycemia (mean HbA1c: 7.9% to 8.1%) despite being on metformin and pioglitazone treatment; the beneficial effects of add-on therapy may be due to glycemic reductions. In our study, patients with T2D had well-controlled HbA1c levels on metformin and pioglitazone (mean baseline HbA1c of 6.6% ± 0.6%), which allowed us to study the impact of SGLT2 inhibitors on NAFLD rather than glycemic control. Thus, in our study, HbA1c decreased by only 0.1% ± 0.2%, but also showed a consistent benefit on abdominal visceral fat and other factors associated with NAFLD.

A second strength of our study is that we assessed various modalities to evaluate changes in body fat depots and the severity of NAFLD. We used both whole-body DXA scans and abdominal fat CT scans to assess fat and muscle depots and transient liver elastography and various equations to assess the severity of NAFLD. Although prior studies in animal models have reported a positive impact of SGLT2 inhibitors on body fat distribution and NAFLD severity [29,30], human studies have reported inconsistent findings. In a 102-week study, dapagliflozin add-on to metformin was associated with a 1.5% reduction in total fat mass as measured by DXA, but was not associated with changes in hepatic lipid contents as measured by magnetic resonance (MR) [31]. Another study, which compared dapagliflozin versus placebo add-on to metformin, failed to find significant changes in MR-measured VFA and SFA [32]. In these studies, however, MR was applied only in a subgroup of patients. Moreover, in the dapagliflozin versus placebo study, there was a significant difference in baseline values between the two groups.

A third strength of our study is that it showed associations between changes in adipose tissue, body weight, NAFLD indexes, and muscle mass. Although the underlying association between weight reduction and liver function improvement in response to treatment with SGLT2 inhibitors has been demonstrated in animal models [29,33], associations between NAFLD severity, weight reduction, and fat distribution have not been fully elucidated in humans. In a Japanese study on the effect of SGLT2 inhibitors on liver function, a reduction in ALT was observed only in the high ALT subgroup (>30 U/L), and this reduction was associated with the baseline ALT level [34]. Despite the presence of normal hepatic function at study enrollment, we demonstrated that ipragliflozin add-on consistently improved NAFLD parameters. Furthermore, improvements in CAP were associated with reductions in body weight. This finding is consistent with a two-pooled phase 3 study in which canagliflozin-related improvements in liver enzymes were associated with reductions in both body weight and HbA1c levels [35]. In that study, changes in liver enzymes were more strongly associated with reductions in HbA1c than with reductions in weight. In contrast, in our study there was no statistically significant difference in HbA1c levels between our two groups, which suggests that ipragliflozin additively reduces visceral fat and ameliorates NAFLD in euglycemic patients with type 2 diabetes.

Our study also has a few limitations. First, although we enrolled ultrasound-confirmed patients with NAFLD, diagnosis was not histologically confirmed. Second, although we encouraged all patients to exercise regularly and avoid overeating, we did not track food or calorie intake across the study, and thus cannot rule out the effect of these factors between the two study groups on the outcomes.

In conclusion, treatment with ipragliflozin significantly ameliorates liver steatosis and excessive fat over 24 weeks in patients with T2D and NAFLD controlled with metformin and pioglitazone. Most weight reduction was due to visceral adipose tissue loss, with significant reductions in liver fat contents. These findings provide further support for the clinical utility of ipragliflozin as an add-on therapy in patients with T2D and NAFLD.

## Figures and Tables

**Figure 1 jcm-09-00259-f001:**
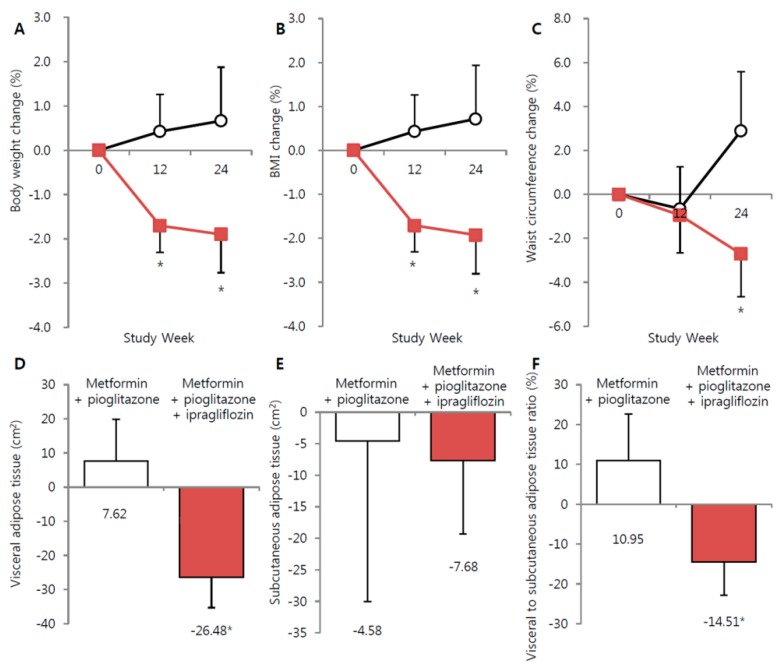
Changes in body weight, body mass index, waist circumference, abdominal adipose tissue, body fat and muscle from baseline to week 24. (**A**) Body weight. (**B**) Body mass index. (**C**) Waist circumference. (**D**) Abdominal visceral adipose tissue assessed by CT. (**E**) Abdominal subcutaneous adipose tissue assessed by CT. (**F**) Abdominal visceral to subcutaneous adipose tissue ratio assessed by CT. Red square: metformin + pioglitazone + ipragliflozin. Black circle: metformin + pioglitazone. * *p* < 0.05.

**Figure 2 jcm-09-00259-f002:**
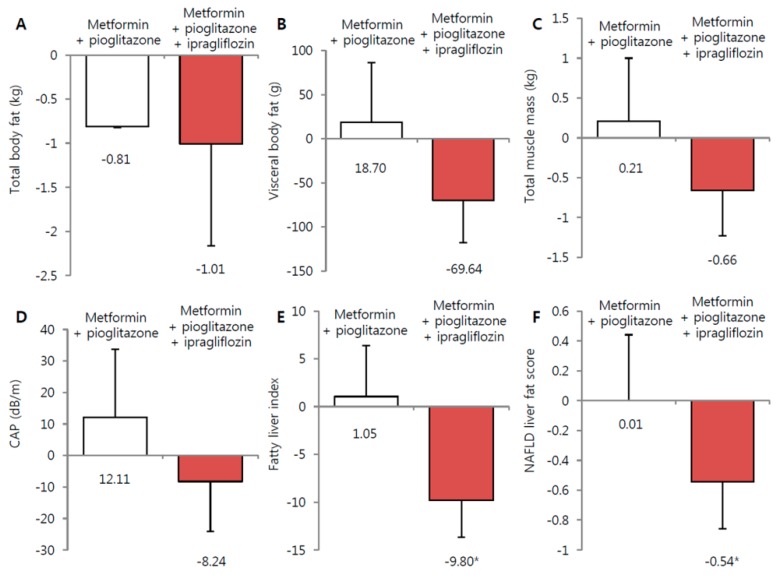
Changes in body fat, muscle and hepatic fat contents from baseline to week 24. (**A**) Total body fat. (**B**) Visceral body fat. (**C**) Total muscle mass by dual-energy x-ray absorptiometry. (**D**) Controlled attenuation parameter assessed by transient liver elastography. (**E**) Fatty liver index. (**F**) Non-alcoholic fatty liver disease liver fat score. Red square: metformin + pioglitazone + ipragliflozin. Black circle: metformin + pioglitazone. * *p* < 0.05.

**Figure 3 jcm-09-00259-f003:**
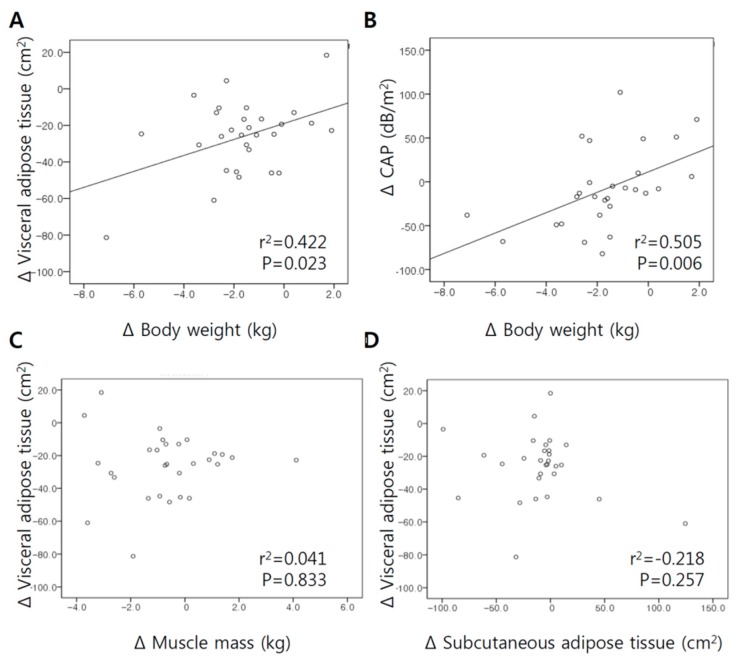
Correlation between changes in body weight, adipose tissue, liver fat and muscle mass. (**A**) Correlation between changes in visceral adipose tissue and body weight. (**B**) Correlation between changes in controlled attenuation parameter and body weight. (**C**) Correlation between changes in visceral adipose tissue and muscle mass. (**D**) Correlation between changes in visceral adipose tissue and subcutaneous adipose tissue.

**Table 1 jcm-09-00259-t001:** Baseline characteristics of the study population.

Parameters	Metformin + Pioglitazone (*n* = 15)	Metformin + Pioglitazone + Ipragliflozin (*n* = 30)	*p*-Value
Age (years)	56.7 ± 11.8	52.5 ± 10.3	0.233
Sex (male), *N* (%)	9 (60.0)	19 (63.3)	>0.999
Diabetes duration (years)	10.1 ± 5.6	9.1 ± 6.0	0.563
Waist circumference (cm)	100.5 ± 7.7	102.1 ± 12.2	0.606
Weight (kg)	81.4 ± 8.5	84.2 ± 16.9	0.471
BMI (kg/m^2^)	30.2 ± 2.5	30.4 ± 5.4	0.871
SBP (mmHg)	124.9 ±9.2	125.3 ± 11.1	0.889
DBP (mmHg)	74.0 ± 9.0	75.6 ± 10.0	0.605
HTN, *N* (%)	11 (73.3)	18 (60.0)	0.378
Dyslipidemia, *N* (%)	15 (100.0)	29 (96.7)	>0.999
FPG (mg/dL)	118.4 ± 19.7	120.1 ± 21.8	0.790
HbA1c (%)	6.6 ± 0.6	6.7 ± 0.7	0.592
HbA1c (mmol/mol)	48.2 ± 6.4	49.4 ± 7.5	0.598
HOMA-IR *	3.4 ± 2.8	2.7 ± 1.8	0.351
HOMA-β (%) *	73.5 ± 38.0	69.0 ± 59.4	0.382
Total cholesterol (mg/dL)	170.3 ± 21.7	186.7 ± 34.9	0.110
HDL cholesterol (mg/dL)	49.9 ± 11.9	50.2 ± 13.3	0.960
Triglycerides (mg/dL)	152.5 ± 91.8	161.3 ± 66.3	0.551
LDL cholesterol (mg/dL)	89.8 ± 17.3	104.2 ± 28.8	0.103
WBC (10^3^/μL)	6.9 ± 1.1	7.1 ± 2.0	0.661
Hemoglobin (mg/dL)	13.7 ± 1.5	14.1 ± 1.4	0.408
Platelet (10^9^/L) *	239.5 ± 52.4	257.3 ± 59.1	0.398
Creatinine (mg/dL)	0.8 ± 0.2	0.7 ± 0.2	0.381
eGFR, EPI (mL/min/1.73 m^2^)	94.8 ± 10.1	101.9 ± 11.8	0.053
Uric acid (mg/dL)	5.3 ± 1.3	5.5 ± 1.5	0.692
Albumin (g/dL)	4.3 ± 0.3	4.4 ± 0.2	0.633
γ-GT (IU/L) *	31.8 ± 15.6	40.5 ± 47.0	0.677
AST (IU/L) *	30.4 ± 19.6	26.3 ± 12.9	0.433
ALT (IU/L) *	31.1 ± 13.5	32.9 ± 24.8	0.779
Transient elastography, CAP (dB/m)	307.7 ± 37.0	305.5 ± 39.5	0.858
**NAFLD score**			
Fatty liver index	66.5 ± 18.7	68.2 ± 21.3	0.786
NAFLD liver fat score	0.9 ± 1.4	0.6 ± 1.5	0.518
**Abdominal fat CT scan**			
VFA (cm^2^)	223.3 ± 90.8	203.0 ± 55.0	0.528
SFA (cm^2^)	230.1 ± 80.6	213.3 ± 111.0	0.336
Ratio of VFA to SFA (%)	49.0 ± 11.4	46.1 ± 12.2	0.422
**DXA scan**			
Total fat mass (kg)	25.0 ± 7.7	24.7 ± 7.0	0.869
Total fat ratio (%)	29.6 ± 7.7	29.6 ± 6.5	0.857
Estimated visceral fat (g)	664.1 ± 190.5	690.8 ± 169.1	0.600
Total muscle mass (kg)	55.8 ± 9.9	56.6 ± 12.1	0.832

* Log-transformed. Data for continuous variables were expressed as mean ± SD for parametric variables. Abbreviations: ALT, alanine transaminase; AST, aspartate aminotransferase; BMI, body mass index; CAP, controlled attenuation parameter; CT, computed tomography; DBP, diastolic blood pressure; DXA, dual-energy x-ray absorptiometry; eGFR, estimated glomerular filtration rate; EPI, epidemiology collaboration equation; FPG, fasting plasma glucose; HbA1c, glycated hemoglobin; HDL, high-density lipoprotein; HTN, hypertension; γ-GT, gamma glutamyl transferase; HOMA-IR, homeostatic model assessment of insulin resistance; LDL, low-density lipoprotein; NAFLD, non-alcoholic fatty liver disease; SBP, systolic blood pressure; SFA, subcutaneous fat area; VFA, visceral fat area; WBC, white blood cell.

**Table 2 jcm-09-00259-t002:** Changes in glycemic, hepatic, and lipid parameters compared between baseline and week 24 by treatment group.

	Metformin + Pioglitazone (*n* = 15)	Metformin + Pioglitazone + Ipragliflozin (*n* = 29)	*p*-Value
**Total VAT (g)**			
Baseline	664.1 ± 190.5	698.0 ± 167.6	0.547
Week 24	686.4 ± 185.3	626.4 ± 198.9	0.338
Changes from baseline	22.3 ± 40.1	**−71.5 ± 21.5 ***	**0.029**
**Total fat (kg)**			
Baseline	25.0 ± 7.6	25.8 ± 8.0	0.766
Week 24	24.3 ± 5.6	24.7 ± 8.2	0.849
Changes from baseline	0.7 ± 1.3	**−1.0 ± 0.3 ***	0.774
**Total muscle (kg)**			
Baseline	55.8 ± 9.9	56.6 ± 12.1	0.832
Week 24	56.1 ± 1.0	55.9 ± 11.4	0.970
Changes from baseline	0.2 ± 0.3	**−0.8 ± 0.3 ***	0.079
**VFA (cm^2^)**			
Baseline	223.3 ± 90.8	209.1 ± 63.3	0.546
Week 24	230.3 ± 87.6	182.9 ± 63.7	0.046
Changes from baseline	7.0 ± 7.7	**−26.2 ± 3.7 ****	**<0.001**
**SFA (cm^2^)**			
Baseline	230.1 ± 80.6	267.5 ± 115.4	0.269
Week 24	228.7 ± 90.0	258.2 ± 99.7	0.341
Changes from baseline	−1.4 ± 5.0	−9.3 ± 7.2	0.460
**CAP (dB/m)**			
Baseline	307.7 ± 37.0	306.6 ± 39.8	0.928
Week 24	319.5 ± 44.8	298.6 ± 45.2	0.207
Changes from baseline	11.7 ± 12.1	−8.0 ± 8.5	0.182
**Body weight (kg)**			
Baseline	81.4 ± 8.5	84.3 ± 17.2	0.470
Week 12	81.8 ± 8.4	82.8 ± 17.1	0.780
Week 24	81.9 ± 7.6	82.6 ± 16.9	0.854
Changes from baseline	0.4 ± 0.6	**−1.6 ± 0.4 ****	**0.003**
**BMI (kg/m^2^)**			
Baseline	30.2 ± 2.5	30.6 ± 5.3	0.734
Week 12	30.3 ± 2.5	30.1 ± 5.4	0.850
Week 24	30.4 ± 2.6	30.1 ± 5.3	0.745
Changes from baseline	0.2 ± 0.2	**−0.6 ± 0.1 ****	**0.001**
**Waist circumference (cm)**			
Baseline	100.5 ± 7.7	102.4 ± 12.3	0.542
Week 12	99.9 ± 8.4	101.2 ± 11.4	0.698
Week 24	100.0 ± 8.2	99.2 ± 11.6	0.815
Changes from baseline	0.5 ± 0.7	**−3.2 ± 0.8 ***	**0.038**
**SBP (mmHg)**			
Baseline	124.9 ± 9.2	125.8 ± 11.1	0.790
Week 12	121.7 ± 9.6	124.6 ± 9.2	0.337
Week 24	128.1 ± 9.6	125.1 ± 10.8	0.382
Changes from baseline	3.2 ± 2.0	−0.6 ± 2.1	0.242
**DBP (mmHg)**			
Baseline	74.0 ± 9.0	75.5 ± 10.2	0.622
Week 12	72.1 ± 7.8	77.8 ± 9.9	0.062
Week 24	75.0 ± 8.5	77.7 ± 8.5	0.449
Changes from baseline	1.6 ± 2.5	2.1 ± 1.6	0.868
**FPG (mg/dL)**			
Baseline	118.4 ± 19.7	121.2 ± 21.3	0.674
Week 12	140.7 ± 39.2 *	125.5 ± 20.6	0.180
Week 24	116.3 ± 20.9	117.5 ± 19.8	0.846
Changes from baseline	−2.1 ± 8.0	−3.7 ± 3.4	0.835
**HbA1c (%)**			
Baseline	6.6 ± 0.6	6.7 ± 0.7	0.595
Week 12	6.9 ± 0.9	6.4 ± 0.5	0.035
Week 24	6.8 ± 0.7 *	6.5 ± 0.7	0.287
Changes from baseline	0.2 ± 0.1	**−0.1 ± 0.2 ***	0.129
**HOMA-IR**			
Baseline	3.4 ± 2.8	2.7 ± 1.8	0.273
Week 24	3.5 ± 2.5	2.2 ± 1.4	0.050
Changes from baseline	0.1 ± 0.8	**−0.5 ± 0.2 ***	0.400
**HOMA-β**			
Baseline	73.5 ± 38.0	66.6 ± 59.0	0.182
Week 24	112.7 ± 151.1	55.7 ± 41.2	0.039
Changes from baseline	39.1 ± 37.4	−10.9 ± 8.4	0.212
**AST (IU/L)**			
Baseline	30.4 ± 19.6	26.6 ± 13.0	0.485
Week 12	28.5 ± 14.4	27.8 ± 16.8	0.781
Week 24	24.7 ± 10.0	24.3 ± 10.6	0.845
Changes from baseline	5.7 ± 3.2	2.3 ± 1.1	0.323
**ALT (IU/L)**			
Baseline	31.1 ± 13.5	33.4 ± 25.1	0.839
Week 12	27.3 ± 8.8	31.0 ± 20.9	0.781
Week 24	26.5 ± 11.8	25.6 ± 16.9	0.511
Changes from baseline	−4.7 ± 2.9	**−7.8 ± 2.6 ****	0.458
**γ-GT (IU/L)**			
Baseline	31.8 ± 15.6	40.9 ± 47.8	0.671
Week 24	29.9 ± 13.8	29.7 ± 24.6	0.510
Changes from baseline	−1.9 ± 2.0	**−11.2 ± 4.9 ****	0.189
**Total cholesterol (mg/dL)**			
Baseline	170.3 ± 21.7	187.1 ± 35.4	0.109
Week 12	170.9 ± 30.1	183.4 ± 33.2	0.260
Week 24	158.4 ± 23.2	184.6 ± 35.6	0.012
Changes from baseline	−11.9 ± 5.8	−2.5 ± 3.6	0.158
**HDL (mg/dL)**			
Baseline	50.0 ± 11.9	50.7 ± 13.2	0.853
Week 12	51.0 ± 16.2	51.1 ± 12.2	0.742
Week 24	49.2 ± 13.0	52.4 ± 10.6	0.271
Changes from baseline	−0.7 ± 1.0	1.7 ± 1.6	0.302
**Triglyceride (mg/dL)**			
Baseline	152.5 ± 91.8	159.7 ± 66.9	0.612
Week 12	177.0 ± 135.5	166.5 ± 99.1	0.825
Week 24	163.5 ± 106.2	149.0 ± 56.4	0.744
Changes from baseline	11.0 ± 13.9	−10.8 ± 11.6	0.258
**LDL (mg/dL)**			
Baseline	91.3 ± 16.9	102.8 ± 28.9	0.106
Week 12	92.0 ± 19.4	100.8 ± 29.1	0.469
Week 24	82.0 ± 20.8	102.4 ± 29.4	0.024
Changes from baseline	**−9.3 ± 3.7 ****	2.1 ± 2.7	0.130

Data are mean ± SD; * *p* ≤ 0.05 compared with baseline, ** *p* ≤ 0.05 compared with baseline. Abbreviations: ALT, alanine transaminase; AST, aspartate aminotransferase; BMI, body mass index; CAP, controlled attenuation parameter; DBP, diastolic blood pressure; DXA, dual-energy x-ray absorptiometry; FLI, fatty liver index; FPG, fasting plasma glucose; HbA1c, glycated hemoglobin; HDL, high-density lipoprotein; γ-GT, gamma glutamyl transferase; HOMA-IR, homeostatic model assessment of insulin resistance; LDL, low-density lipoprotein; LFS, non-alcoholic fatty liver fat score; NAFLD, non-alcoholic fatty liver disease; SBP, systolic blood pressure; SFA, subcutaneous fat area; VFA, visceral fat area.

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
