# Peer review of "Ipragliflozin Additively Ameliorates Non-Alcoholic Fatty Liver Disease in Patients with Type 2 Diabetes Controlled with Metformin and Pioglitazone: A 24-Week Randomized Controlled Trial"

_jcm, 2020, doi:10.3390/jcm9010259_

Round 1

Reviewer 1 Report

In this manuscript, the authors tried to assess the additive effect of ipragliflozin on NAFLD in patients with type 2 diabetes treated with metformin and pioglitazone. They concluded that treatment with ipragliflozin significantly ameliorates liver steatosis and excessive fat over 24 weeks and most weight reduction was due to visceral adipose tissue loss with significant reductions in liver fat contents. In total, this is an innovative and well-written article. However, there are some problems that the authors need to address.

1 How was the treatment about NAFLD including ursodeoxycholic acid or Vitamin E between the two groups?

2 What is the result of multivariable-adjusted logistic regression analyses performed to test the independent association between ipragliflozin response and other clinical factors?

3 I think that Figure 2 and Figure S2 are same.

Author Response

First of all, we would like to thank the Academic Editor for his/her favorable comments and criticism. We have responded to the following points with consideration and hope that our changes will improve the quality of the article.

Comment 1. How was the treatment about NAFLD including ursodeoxycholic acid or Vitamin E between the two groups?

Response 1. Thank you for the important suggestion. None of the study subject were treated with UDCA, vitamin E, or other medication which affect NAFLD.

Comment 2. What is the result of multivariable-adjusted logistic regression analyses performed to test the independent association between ipragliflozin response and other clinical factors?

Response 2. We identified the ipragliflozin responders as subjects with body weight reduction (1.6kg, median value). Although we could not find any statistically significant factors associated with ipragliflozin response in multivariable-adjusted logistic regression analyses, ipragliflozin response group tended to have higher baseline CAP and muscle mass. We newly described this in the result section as follows: 

We identified the ipragliflozin responders as subjects with body weight reduction (1.6kg, median value). Ipragliflozin response group tended to have higher baseline CAP and muscle mass, however, there was no clinical factors associated to ipragliflozin response in multivariable-adjusted logistic regression analyses.

Comment 3. I think that Figure 2 and Figure S2 are same.

Response 3. There was a mistake indicating the figures. We deleted Figure S2.

Reviewer 2 Report

I have read the manuscript: Ipragliflozin additively ameliorates non-alcoholic fatty liver disease in patients with type 2 diabetes controlled with metformin and pioglitazone: A 24-week randomized controlled trial with pleasure and great interest.

This is a high-quality scientific text, very well prepared and very clearly presented. I have no hesitations, remarks or questions except a few details:

The supplementary materials were not available for the review.

Results

2.2. Laboratory and imaging studies
Please, briefly describe the methodology of the controlled attenuation parameter (CAP), as described previously [5, 20]. We also analyzed each patient’s fatty liver index [21], and NAFLD liver fat score [22].

The rearranging the tables into 3 lines tables, and bolding statistically significant values would help the readers to analyze the data better.

In table 1 NAFLD score should be bolded.

Author Response

Comment 1. Please, briefly describe the methodology of the controlled attenuation parameter (CAP), as described previously [5, 20]. We also analyzed each patient’s fatty liver index [21], and NAFLD liver fat score [22].

Response 1. Thank you for the important suggestion. We newly described as follows: Briefly, the CAP measures ultrasonic attenuations at 3.5 MHz using signals acquired by transient liver elastography (FibroScan®; Echosens, Paris, France). The final CAP value was the median of each individual CAP values using the same valid measurements.

The equations are described in Table S1.

Table S1. Noninvasive calculation methods for assessing liver steatosis

Models

Equations

Fatty liver index

(e 0.953×loge (triglycerides) + 0.139*BMI + 0.718×loge (GGT) + 0.053×waist circumference - 15.745) / (1 + e 0.953×loge (triglycerides) + 0.139×BMI + 0.718×loge (GGT) + 0.053×waist circumference - 15.745) × 100

NAFLD liver fat score

 - 2.89 + 1.18 × metabolic syndrome (yes = 1/no = 0) + 0.45 × diabetes (yes = 2/no = 0) + 0.15 × (fasting insulin, μU/L) + 0.04 × AST + 0.94 × AST/ALT ratio

NAFLD, non-alcoholic fatty liver disease; BMI, body mass index; AST, aspartate aminotransferase; ALT, alanine transaminase; WC, waist circumference; HDL, high-density lipoprotein; GGT, γ-glutamyl transpeptidase.

Comment 2. The rearranging the tables into 3 lines tables, and bolding statistically significant values would help the readers to analyze the data better.

Response 2. We newly rearranged Table 1 into 3 lines table, and bolding statistically significant values in Table 2. 

Table 1. Baseline characteristics of the study population.

Parameters

Metformin + pioglitazone (n=15)

Metformin + pioglitazone + ipragliflozin (n=30)

P value

Age (years)

56.7 ± 11.8

52.5 ± 10.3

0.233

Sex (male), N (%)

9 (60.0)

19 (63.3)

>0.999

Diabetes duration (years)

10.1 ± 5.6

9.1 ± 6.0

0.563

Waist circumference (cm)

100.5 ± 7.7

102.1 ± 12.2

0.606

Weight (kg)

81.4 ± 8.5

84.2 ± 16.9

0.471

BMI (kg/m2)

30.2 ± 2.5

30.4 ± 5.4

0.871

SBP (mmHg)

124.9 ±9.2

125.3 ± 11.1

0.889

DBP (mmHg)

74.0 ± 9.0

75.6 ± 10.0

0.605

HTN, N (%)

11 (73.3)

18 (60.0)

0.378

Dyslipidemia, N (%)

15 (100.0)

29 (96.7)

>0.999

FPG (mg/dL)

118.4 ± 19.7

120.1 ± 21.8

0.790

HbA1c (%)

6.6 ± 0.6

6.7 ± 0.7

0.592

HbA1c (mmol/mol)

48.2 ± 6.4

49.4 ± 7.5

0.598

HOMA-IR*

3.4 ± 2.8

2.7 ± 1.8

0.351

HOMA-β (%)*

73.5 ± 38.0

69.0 ± 59.4

0.382

Total cholesterol (mg/dL)

170.3 ± 21.7

186.7 ± 34.9

0.110

HDL cholesterol (mg/dL)

49.9 ± 11.9

50.2 ± 13.3

0.960

Triglycerides (mg/dL)

152.5 ± 91.8

161.3 ± 66.3

0.551

LDL cholesterol (mg/dL)

89.8 ± 17.3

104.2 ± 28.8

0.103

WBC (103/μL)

6.9 ± 1.1

7.1 ± 2.0

0.661

Hemoglobin (mg/dL)

13.7 ± 1.5

14.1 ± 1.4

0.408

Platelet (109/L)*

239.5 ± 52.4

257.3 ± 59.1

0.398

Creatinine (mg/dL)

0.8 ± 0.2

0.7 ± 0.2

0.381

eGFR, EPI (mL/min/1.73 m2)

94.8 ± 10.1

101.9 ± 11.8

0.053

Uric acid (mg/dL)

5.3 ± 1.3

5.5 ± 1.5

0.692

Albumin (g/dL)

4.3 ± 0.3

4.4 ± 0.2

0.633

γ-GT (IU/L)*

31.8 ± 15.6

40.5 ± 47.0

0.677

AST (IU/L)*

30.4 ± 19.6

26.3 ± 12.9

0.433

ALT (IU/L)*

31.1 ± 13.5

32.9 ± 24.8

0.779

Transient elastography, CAP (dB/m)

307.7 ± 37.0

305.5 ± 39.5

0.858

NAFLD score

 Fatty liver index

66.5 ± 18.7

68.2 ± 21.3

0.786

 NAFLD liver fat score

0.9 ± 1.4

0.6 ± 1.5

0.518

Abdominal fat CT scan

VFA (cm2)

223.3 ± 90.8

203.0 ± 55.0

0.528

SFA (cm2)

230.1 ± 80.6

213.3 ± 111.0

0.336

Ratio of VFA to SFA (%)

49.0 ± 11.4

46.1 ± 12.2

0.422

DXA scan

Total fat mass (kg)

25.0 ± 7.7

24.7 ± 7.0

0.869

Total fat ratio (%)

29.6 ± 7.7

29.6 ± 6.5

0.857

Estimated visceral fat (g)

664.1 ± 190.5

690.8 ± 169.1

0.600

Total muscle mass (kg)

55.8 ± 9.9

56.6 ± 12.1

0.832

*log-transformed. Data for continuous variables were expressed as mean ± SD for parametric variables. Abbreviation: ALT, alanine transaminase; AST, aspartate aminotransferase; BMI, body mass index; CAP, controlled attenuation parameter; CT, computed tomography; DBP, diastolic blood pressure; DXA, dual-energy x-ray absorptiometry; eGFR, estimated glomerular filtration rate; EPI, Epidemiology Collaboration equation; FPG, fasting plasma glucose; HDL, high-density lipoprotein; HTN, hypertension; γ-GT, gamma glutamyl transferase; HOMA-IR, homeostatic model assessment of insulin resistance; LDL, low-density lipoprotein; NAFLD, nonalcoholic fatty liver disease; SBP, systolic blood pressure; SFA, subcutaneous fat area; VFA, visceral fat area; WBC, white blood cell.

Table 2. Changes in glycemic, hepatic, and lipid parameters compared between baseline and week 24 by treatment group.

Metformin + pioglitazone

(n=15)

Metformin + pioglitazone

+ ipragliflozin (n=29)

p value

Total VAT (g)

 Baseline

664.1 ± 190.5

698.0 ± 167.6

0.547

 Week 24

686.4 ± 185.3

626.4 ± 198.9

0.338

 Changes from baseline

22.3 ± 40.1

-71.5 ± 21.5*

0.029

Total fat (kg)

 Baseline

25.0 ± 7.6

25.8 ± 8.0

0.766

 Week 24

24.3 ± 5.6

24.7 ± 8.2

0.849

 Changes from baseline

0.7 ± 1.3

-1.0 ± 0.3*

0.774

Total muscle (kg)

 Baseline

55.8 ± 9.9

56.6 ± 12.1

0.832

 Week 24

56.1 ± 1.0

55.9 ± 11.4

0.970

 Changes from baseline

0.2 ± 0.3

-0.8 ± 0.3*

0.079

VFA (cm2)

 Baseline

223.3 ± 90.8

209.1 ± 63.3

0.546

 Week 24

230.3 ± 87.6

182.9 ± 63.7

0.046

 Changes from baseline

7.0 ± 7.7

-26.2 ± 3.7**

<0.001

SFA (cm2)

 Baseline

230.1±80.6

267.5 ± 115.4

0.269

 Week 24

228.7±90.0

258.2 ± 99.7

0.341

 Changes from baseline

-1.4±5.0

-9.3 ± 7.2

0.460

CAP (dB/m)

 Baseline

307.7 ± 37.0

306.6 ± 39.8

0.928

 Week 24

319.5 ± 44.8

298.6 ± 45.2

0.207

 Changes from baseline

11.7 ± 12.1

-8.0 ± 8.5

0.182

Body weight (kg)

 Baseline

81.4 ± 8.5

84.3 ± 17.2

0.470

 Week 12

81.8 ± 8.4

82.8 ± 17.1

0.780

 Week 24

81.9 ± 7.6

82.6 ± 16.9

0.854

 Changes from baseline

0.4 ± 0.6

-1.6 ± 0.4**

0.003

BMI (kg/m2)

 Baseline

30.2 ± 2.5

30.6 ± 5.3

0.734

 Week 12

30.3 ± 2.5

30.1 ± 5.4

0.850

 Week 24

30.4 ± 2.6

30.1 ± 5.3

0.745

 Changes from baseline

0.2 ± 0.2

-0.6 ± 0.1**

0.001

Waist circumference (cm)

 Baseline

100.5 ± 7.7

102.4 ± 12.3

0.542

 Week 12

99.9 ± 8.4

101.2 ± 11.4

0.698

 Week 24

100.0 ± 8.2

99.2 ± 11.6

0.815

 Changes from baseline

0.5 ± 0.7

-3.2 ± 0.8*

0.038

SBP (mmHg)

 Baseline

124.9 ± 9.2

125.8 ± 11.1

0.790

 Week 12

121.7 ± 9.6

124.6 ± 9.2

0.337

 Week 24

128.1 ± 9.6

125.1 ± 10.8

0.382

 Changes from baseline

3.2 ± 2.0

-0.6 ± 2.1

0.242

DBP (mmHg)

 Baseline

74.0 ± 9.0

75.5 ± 10.2

0.622

 Week 12

72.1 ± 7.8

77.8 ± 9.9

0.062

 Week 24

75.0 ± 8.5

77.7 ± 8.5

0.449

 Changes from baseline

1.6 ± 2.5

2.1 ± 1.6

0.868

FPG (mg/dL)

 Baseline

118.4 ± 19.7

121.2 ± 21.3

0.674

 Week 12

140.7 ± 39.2*

125.5 ± 20.6

0.180

 Week 24

116.3 ± 20.9

117.5 ± 19.8

0.846

 Changes from baseline

-2.1 ± 8.0

-3.7 ± 3.4

0.835

HbA1c (%)

 Baseline

6.6 ± 0.6

6.7 ± 0.7

0.595

 Week 12

6.9 ± 0.9

6.4 ± 0.5

0.035

 Week 24

6.8 ± 0.7*

6.5 ± 0.7

0.287

 Changes from baseline

0.2 ± 0.1

-0.1 ± 0.2*

0.129

HOMA-IR

 Baseline

3.4 ± 2.8

2.7 ± 1.8

0.273

 Week 24

3.5 ± 2.5

2.2 ± 1.4

0.050

 Changes from baseline

0.1 ± 0.8

-0.5 ± 0.2*

0.400

HOMA-β

 Baseline

73.5 ± 38.0

66.6 ± 59.0

0.182

 Week 24

112.7 ± 151.1

55.7 ± 41.2

0.039

 Changes from baseline

39.1 ± 37.4

-10.9 ± 8.4

0.212

AST (IU/L)

 Baseline

30.4 ± 19.6

26.6 ± 13.0

0.485

 Week 12

28.5 ± 14.4

27.8 ± 16.8

0.781

 Week 24

24.7 ± 10.0

24.3 ± 10.6

0.845

 Changes from baseline

5.7 ± 3.2

2.3 ± 1.1

0.323

ALT (IU/L)

 Baseline

31.1 ± 13.5

33.4 ± 25.1

0.839

 Week 12

27.3 ± 8.8

31.0 ± 20.9

0.781

 Week 24

26.5 ± 11.8

25.6 ± 16.9

0.511

 Changes from baseline

-4.7 ± 2.9

-7.8 ± 2.6**

0.458

γ-GT (IU/L)

 Baseline

31.8 ± 15.6

40.9 ± 47.8

0.671

 Week 24

29.9 ± 13.8

29.7 ± 24.6

0.510

 Changes from baseline

-1.9 ± 2.0

-11.2 ± 4.9**

0.189

Total cholesterol (mg/dL)

 Baseline

170.3 ± 21.7

187.1 ± 35.4

0.109

 Week 12

170.9 ± 30.1

183.4 ± 33.2

0.260

 Week 24

158.4 ± 23.2

184.6 ± 35.6

0.012

 Changes from baseline

-11.9 ± 5.8

-2.5 ± 3.6

0.158

HDL (mg/dL)

 Baseline

50.0 ± 11.9

50.7 ± 13.2

0.853

 Week 12

51.0 ± 16.2

51.1 ± 12.2

0.742

 Week 24

49.2 ± 13.0

52.4 ± 10.6

0.271

 Changes from baseline

-0.7 ± 1.0

1.7 ± 1.6

0.302

Triglyceride (mg/dL)

 Baseline

152.5 ± 91.8

159.7 ± 66.9

0.612

 Week 12

177.0 ± 135.5

166.5 ± 99.1

0.825

 Week 24

163.5 ± 106.2

149.0 ± 56.4

0.744

 Changes from baseline

11.0 ± 13.9

-10.8 ± 11.6

0.258

LDL (mg/dL)

 Baseline

91.3 ± 16.9

102.8 ± 28.9

0.106

 Week 12

92.0 ± 19.4

100.8 ± 29.1

0.469

 Week 24

82.0 ± 20.8

102.4 ± 29.4

0.024

 Changes from baseline

-9.3 ± 3.7**

2.1 ± 2.7

0.130

Data are mean ± SD. * P ≤0.05 compared with baseline, ** P ≤0.05 compared with baseline. Abbreviation: ALT, alanine transaminase; AST, aspartate aminotransferase; BMI, body mass index; CAP, controlled attenuation parameter; DBP, diastolic blood pressure; DXA, dual-energy x-ray absorptiometry; FLI, fatty liver index; FPG, fasting plasma glucose; HDL, high-density lipoprotein; γ-GT, gamma glutamyl transferase; HOMA-IR, homeostatic model assessment of insulin resistance; LDL, low-density lipoprotein; LFS, nonalcoholic fatty liver fat score; NAFLD, nonalcoholic fatty liver disease; SBP, systolic blood pressure; SFA, subcutaneous fat area; VFA, visceral fat area.

Comment 3. In table 1 NAFLD score should be bolded.

Response 3. We highlighted NAFLD score in bold type.